# Single-Dose SDA-Rich Echium Oil Increases Plasma EPA, DPAn3, and DHA Concentrations

**DOI:** 10.3390/nu11102346

**Published:** 2019-10-02

**Authors:** Theresa Greupner, Elisabeth Koch, Laura Kutzner, Andreas Hahn, Nils Helge Schebb, Jan Philipp Schuchardt

**Affiliations:** 1Institute of Food Science and Human Nutrition, Leibniz University Hannover, 30177 Hannover, Germany; 2Chair of Food Chemistry, Faculty of Mathematics and Natural Sciences, University of Wuppertal, 42119 Wuppertal, Germany

**Keywords:** polyunsaturated fatty acid metabolism, conversion, stearidonic acid

## Abstract

The omega-3 (*n*3) polyunsaturated fatty acids (PUFA) eicosapentaenoic acid (EPA) and docosahexaenoic acid (DHA) are associated with health benefits. The primary dietary source of EPA and DHA is seafood. Alpha-linoleic acid (ALA) has not been shown to be a good source for EPA and DHA; however, stearidonic acid (SDA)—which is naturally contained in echium oil (EO)—may be a more promising alternative. This study was aimed at investigating the short-term *n*3 PUFA metabolism after the ingestion of a single dose of EO. Healthy young male subjects (*n* = 12) ingested a single dose of 26 g of EO after overnight fasting. Plasma fatty acid concentrations and relative amounts were determined at baseline and 2, 4, 6, 8, 24, 48, and 72 h after the ingestion of EO. During the whole examination period, the participants received standardized nutrition. Plasma ALA and SDA concentrations increased rapidly after the single dose of EO. Additionally, EPA and DPA*n*3 concentrations both increased significantly by 47% after 72 h compared to baseline; DHA concentrations also significantly increased by 21% after 72 h. To conclude, EO increases plasma ALA, SDA, EPA, DPA*n*3, and DHA concentrations and may be an alternative source for these *n*3 PUFAs.

## 1. Introduction

Long-chain (LC) omega-3 (*n*3) polyunsaturated fatty acids (PUFAs), namely eicosapentaenoic acid (EPA, C20:5*n*3) and docosahexaenoic acid (DHA, C22:6*n*3), are associated with health benefits including lower risk for cardiovascular disease and mortality [1,2,3,4,5], reduction of inflammatory conditions [2,6], angiogenesis, tumor growth, metastasis [2,7], and better visual and neurological development [8]. More and more, the positive health effects of docosapentaenoic acid (DPA*n*3, C22:5*n*3) are coming into focus [9,10,11]. However, seafood intake—the main dietary source of EPA and DHA—and blood status of EPA and DHA are low in most countries worldwide [12,13]. Therefore, alternative dietary sources of LC *n*3 PUFAs are necessary in order to fill the existing intake gap.

The essential plant-derived *n*3 PUFA alpha-linolenic acid (ALA, C18:3*n*3) can be converted into EPA, DPA*n*3, and theoretically also to DHA in a multistep elongation and desaturation reaction [14]. However, studies have shown that the conversion of ALA to EPA and especially to DHA is very inefficient or non-existent in humans as reviewed in Reference [15]. Due to the concurrent high linoleic acid (LA, C18:2*n*6) intake—which is typical in many Western diets—even large doses of ALA result in comparatively small increases in plasma EPA and no or negative effects on DHA [16,17,18]. A 12-week linseed oil (LO) supplementation study with a comparatively high ALA dose showed that DHA concentrations in red blood cells decreased significantly [17].

Another more promising source for a higher conversion of precursor *n*3 PUFAs to LC *n*3 PUFAs might be echium oil (EO)—one of a few natural sources of stearidonic acid (SDA, C18:4*n*3). SDA is an intermediate in the conversion of ALA to EPA, more precisely the product of ALA desaturation by delta-6-desaturase (D6D), whose activity is regulated by the expression of the fatty acid desaturase 2 (FADS2) gene. As D6D is the rate limiting enzyme in the conversion of ALA to EPA [19,20], EPA synthesis from SDA might be more efficient than from ALA [21].

Previous studies have shown that supplementation with SDA-rich oils resulted in higher EPA levels compared to ALA-rich oils, while neither ALA- nor SDA-rich oils had an effect on DHA levels [22,23,24,25] as reviewed in Reference [26]. Short-term changes in the plasma fatty acid pattern after a single dose of EO would give a deeper insight into the PUFA metabolism. Therefore, this study aimed at investigating the short-term *n*3 PUFA metabolism after the ingestion of a single dose of 26 g of EO over a period of 72 h under standardized conditions/nutrition.

## 2. Materials and Methods

This investigator-initiated study was conducted according to the guidelines laid down in the Declaration of Helsinki, and all procedures involving human subjects were approved by the ethic committee at the medical chamber of Lower Saxony (Hannover, Germany). Written informed consent was obtained from all subjects. The study is registered in the German clinical trial register (no. DRKS00012257) and was conducted at the Institute of Food Science and Human Nutrition, Leibniz University Hannover, Germany.

Briefly, the study consisted of a screening, a 4-week run-in phase, a single-dose EO ingestion, and a 72-h lasting examination under standardized nutrition. Recruitment, screening procedure, and inclusion and exclusion criteria can be found in the SI. During the run-in phase, participants were requested to abstain from fish, seafood, and ALA-rich vegetable oils, such as LO or chia seeds to minimize nutritional effects on the plasma fatty acid pattern. The examination started with a fasting blood draw, the determination of blood pressure, body height, body weight, and heart rate, and the subjects completed a questionnaire to obtain information about changes in medication, diet, and lifestyle habits (e.g. physical activity) compared to the screening questionnaire. Subsequently, 26 g of EO (Table 1) was ingested. Possible side effects of the EO were recorded 2 h after the single dose ingestion with a questionnaire. Further blood samples were taken at 2, 4, 6, 8, 24, 48, and 72 h after the ingestion of the test oil. The standardized nutrition started with the lunch meal one day prior to the baseline blood collection and ended with the last blood draw after 72 h. The diet was poor in PUFAs and portion sizes were adjusted to energy demands (small and large portion size) of the participants. Volunteers were allowed to drink water, tea and coffee (without milk/sugar) *ad libitum*. Macronutrient and fatty acid composition of the standardized nutrition can be found in Appendix A.

Blood samples were obtained by venipuncture of an arm vein using Multifly needles (Sarstedt, Nümbrecht, Germany) into serum and EDTA monovettes (Sarstedt). For analysis of fatty acids in plasma, EDTA blood monovettes were centrifuged for 10 minutes at 1500 × g and 4 °C and plasma was transferred into 1.5 mL plastic tubes (Sarstedt), where it was immediately frozen and stored at −80 °C until extraction and analysis. All transfer steps were carried out on ice. Serum lipid levels, liver enzymes, and small blood count at baseline were determined in the LADR laboratory (Laborärztliche Arbeitsgemeinschaft für Diagnostik und Rationalisierung e.V.), Hannover, Germany.

Fatty acid pattern of EO (*n* = 3) as well as plasma fatty acid concentrations were determined by means of gas chromatography (GC) with flame ionization detection (FID) as described in Reference [27] with slight modifications. In brief, lipids from blood plasma were extracted with methyl *tert*-butyl ether and methanol and derivatized with methanolic hydrogen chloride. The resulting fatty acid methyl esters (FAME) were quantified using FAME C25:0 as internal standard. In addition to the determined concentration, expressed as µg fatty acid per mL plasma, the relative amount (% of total fatty acids) of each fatty acid was calculated based on peak areas. EO was diluted with *n*-hexane and directly derivatized with methanolic hydrogen chloride. Based on the determined relative fatty acid pattern, the concentration of fatty acids in g/100 g EO was calculated assuming that the fatty acids in EO are primarily bound in triglycerides.

Fatty acid concentrations in plasma, anthropometrical measures, liver enzymes, and serum lipid levels are shown as mean ± standard deviation (SD). If the plasma concentration of a fatty acid was below the lower limit of quantification (LLOQ) in a maximum of 50% of the samples, the LLOQ (e.g., 0.5 µg/ml) was inserted to calculate mean and SD for this respective fatty acid. If the plasma concentration of a fatty acid could not be quantified in more than 50% of the samples, the LLOQ was given instead of mean ± SD, and no statistical evaluation was performed for this parameter. The distributions of the plasma fatty acid data were analyzed by the Kolmogorov–Smirnov test. Statistical differences between the time points were tested with ANOVA with repeated measurements for parametric data, followed by post-hoc t-tests for paired samples with Holm–Bonferroni-adjusted levels of significance and for non-parametric data with Friedman test followed by Dunn–Bonferroni post-hoc tests. Statistical tests were only performed for analytes that were quantified in the study population at all time points. Statistical significance was set at p ≤ 0.05 for all analyses. All statistical analyses were carried out with SPSS software (Version 24, SPSS Inc., Chicago, IL, USA).

## 3. Results

### 3.1. Study Population

Twelve male subjects were included in the study and completed the examination (no drop-outs). The participants (mean age 24.6 ± 2.43 years) were healthy and had a normal BMI (24.6 ± 2.06 kg/m²) (Table 2). All subjects were omnivores with low fish consumption (≤1 serving fish per week). Liver enzymes and serum lipid profile (Table 2) were in a normal range at baseline. EO contained 11.8 g/100 g of SDA and 30.2 g/100 g of ALA (Table 1), hence, the single dose of 26 g delivered 3.1 g of SDA and 7.9 g of ALA. 

Of the 12 subjects, 6 reported side effects 2 h after the single dose ingestion of EO. The following symptoms were mentioned (multiple answers were possible): abdominal pain (1×), belching (2×), heartburn (1×), fishy taste (2×), flatulence (1×), diarrhea (2×), and nausea (1×).

### 3.2. Changes of Plasma Fatty Acid Concentrations

The plasma concentrations of both *n*3 PUFAs ALA and SDA over time resemble a classic inverted U-shape curve with an initial rapid increase. ALA concentrations increased significantly compared to baseline (15.2 ± 4.96 µg/ml) after 2 (45.8 ± 30.5 µg/ml), 4 (72.2 ± 60.7 µg/ml), 6 (123 ± 69.4 µg/ml), and 8 h (72.1 ± 29.7 µg/ml) and decreased again to concentrations that were not significantly different compared to baseline after 24 (28.7 ± 8.77 µg/ml), 48 (14.7 ± 3.97 µg/ml), and 72 h (19.6 ± 5.70 µg/ml) (Figure 1A). At baseline, SDA plasma concentrations were below the LLOQ of 0.5 µg/ml in more than 50% of the samples. The single dose of EO led to a strong increase of SDA concentrations after 2 (12.3 ± 11.6 µg/ml), 4 (21.2 ± 22.2 µg/ml), 6 (43.6 ± 29.4 µg/ml), and 8 h (19.1 ± 12.0 µg/ml) and decreased again after 24 (3.05 ± 1.17 µg/ml), 48 (1.18 ± 0.47 µg/ml), and 72 h (1.04 ± 0.52 µg/ml) (Figure 1B). The trend of ALA and SDA was similarly pronounced with respect to the relative fatty acid pattern (Figure 1A,B). EPA concentrations increased with a time lag and were significantly different to baseline (15.2 ± 6.57 µg/ml) after 6 (18.2 ± 7.41 µg/ml), 8 (21.4 ± 8.19 µg/ml), 24 (22.5 ± 8.68 µg/ml), and 72 h (24.8 ± 8.33 µg/ml). Regarding relative amounts of EPA, a significant increase compared to baseline (0.52 ± 0.18%) could be observed after 24 (0.86 ± 0.27%), 48 (0.85 ± 0.25%), and 72 h (0.85 ± 0.23%) (Figure 1C). Plasma concentrations of DPA*n*3 increased slightly after the single dose of EO and even reached statistical significance compared to baseline (16.4 ± 6.27 µg/ml) after 72 (24.1 ± 4.89 µg/ml) h (Figure 1D). Additionally, DHA plasma concentrations increased slightly (Figure 1E); however, the increase was not linear. Although DHA concentrations increased significantly after 8 (48.2 ± 15.1 µg/ml) and 72 h (48.7 ± 13.3 µg/ml) compared to baseline (40.3 ± 12.4 µg/ml), they were significantly lower compared to baseline after 48 h (33.1 ± 9.84 µg/ml). Taking into account the total fatty acids in plasma, relative amounts of DPA*n*3 showed a slight increase at later time points, while DHA was marginally elevated only after 72 h. Especially due to the strong ALA and SDA increase, the sum of (Σ) *n*3 PUFA also increased significantly in response to the single dose of EO. The increase was significant compared to baseline (90.4 ± 28.7 µg/ml) after 6 (257 ± 114 µg/ml), 8 (196 ± 64.3 µg/ml), 24 (121 ± 34.0 µg/ml), and 72 h (125 ± 31.6 µg/ml) (Figure 1F). No changes were observed for C20:4*n*6 or Σ*n*6 PUFA. The complete plasma fatty acid patterns of all measurement time points can be found in Appendix A.

## 4. Discussion

The present study assessed short-term changes in plasma fatty acid concentrations after a single dose of EO. A focus was set on changes of *n*3 PUFA concentrations.

Plasma is among the three most commonly used blood sample types for the investigation of the PUFA status, and as plasma fatty acid concentrations are altered by prior meal consumption [28], it might be the most suitable medium for the investigation of the short-term metabolism of LC *n*3 PUFAs. Nonetheless, differences in the fatty acid—especially *n*3 PUFA—intake from the background diet of individual subjects would lead to biased results. To overcome these obstacles, the diet of the participants was standardized during the whole examination, including two meals on the day before the baseline measurement, to prevent bias. Additionally, only omnivores with low fish consumption were included to ensure a homogeneous study collective. This careful selection has led to lower standard deviations or standard errors of the mean of the baseline plasma *n*3 PUFA pattern compared to other studies [29,30].

Moreover, the metabolism of PUFAs has a variety of influencing factors, like age [31,32,33,34], sex [35,36,37], BMI [31,38,39], diet [31,36,40], and smoking status [33,38]. These factors lead to variability of data. Consequently, to minimize this variability, a homogenous collective of healthy, non-smoking men within a narrow range regarding age (mean age 24.6 ± 2.43 years) and BMI (24.6 ± 2.06 kg/m^2^) was chosen to investigate the effect of a single EO dose on *n*3 PUFA concentrations in plasma.

As expected, plasma concentrations of ALA and SDA increased rapidly as a consequence of EO intake. EPA, DPA*n*3, and DHA, the direct metabolites of ALA and SDA, also increased significantly. Although the increase of EPA was more pronounced and significant compared to baseline after 6, 8, 24, and 72 h, the DPA*n*3 and DHA increase was marginal and only significant after 72 h and for DHA additionally after 8 h. The effect of a single dose of EO or LO on the plasma PUFA pattern has not been investigated before. Only a few studies measured PUFA (plasma) patterns after daily supplementation of SDA-rich oils over 3 to 16 weeks (reviewed in Reference [26]). Those studies uniquely observed an increase in EPA and DPA*n*3, but no effect on DHA in different blood fractions was observed [22,24,41,42,43,44,45,46,47]. In a direct comparison between eight-week supplementation of EO and LO, EO was more efficient compared to LO in increasing EPA and DPA*n*3 [48], possibly due to the presence of SDA. This can be explained by the circumvention of the D6D, which is the rate limiting enzyme in the conversion process [49,50]. It is utilized twice in the elongation and desaturation process of *n*3 PUFAs: it converts ALA to SDA and, with lower affinity than the first conversion step, also 24:5*n*3 to 24:6*n*3. Through a direct intake of SDA, the first rate-limiting step is already bypassed; therefore, the increase of EPA and DPA*n*3 is higher.

Surprisingly, EO and LO supplementation both led to a slight DHA decrease in a previous study [48]. DHA is synthesized from DPA*n*3 by elongation of DPA*n*3 to 24:5*n*3, a subsequent desaturation by D6D to 24:6*n*3, and a final β-oxidation in the peroxisome [51]. Due to (1) the lower affinity of D6D to 24:5*n*3 compared to ALA and (2) the high abundance of ALA through the intake via EO, causing competitive inhibition of D6D [52], the conversion of ALA as well as SDA to DHA is (generally) estimated to be very low or nonexistent [26]. Moreover, both, a medium-term (seven days) EPA and DPA*n*3 supplementation (dose: 2 g for the first day and 1 g for the subsequent six days) did not lead to an increase of DHA plasma phospholipids [53]. This further underlines the low capacity of D6D for the desaturation of 24:5*n*3 to 24:6*n*3, which is necessary for the formation of DHA.

Our results show slightly increased DHA concentrations, which was significant after 8 and 72 h following single dose EO intake. However, the DHA concentrations were lower compared to baseline after 48 h. It should be noted that the relative amount of DHA seems less affected by the EO intervention; therefore, the share of DHA with respect to total fatty acids is mostly stable over the study period. Hence, no constant DHA increase could be observed. If this short-term increase has (1) any physiological significance and (2) stands in contrast to longer-term SDA supplementation studies showing no effect on DHA the effect of EO supplementation on the DHA status ought to be investigated in future trials. Moreover, it would be interesting to directly compare the plasma fatty acid patterns after the ingestion of single doses of EO and LO. The missing control group is the main limitation of this study. Further, the study is limited by a small sample size of exclusively male subjects. It is questionable whether these results can be reproduced with female subjects as sex differences in the efficiency of the conversion of ALA to DHA have been previously shown [54,55].

Half of the participants stated that they had side effects after the ingestion of the EO. However, due to the missing control group it is not clear if these side effects are possibly in part attributable to the study conditions—especially the standardized diet. Nonetheless, the high single dose of EO is apparently not well tolerated by half of the study population. 

## 5. Conclusions

The present study showed that a single dose of EO results in a significant increase of plasma EPA, DPA*n*3, and DHA concentrations after 72 h beside an increase of ALA and SDA. Therefore, SDA from EO may be a more promising alternative plant source for EPA, DPA*n*3, and DHA compared to ALA from LO due to its effect on DHA. 

## Figures and Tables

**Figure 1 nutrients-11-02346-f001:**
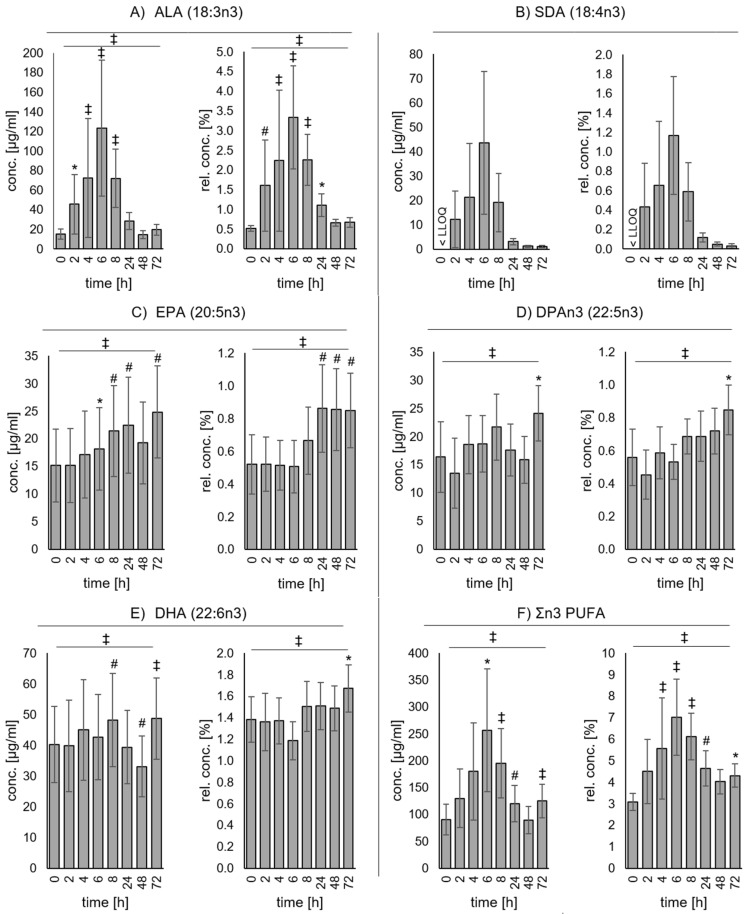
Concentration of selected omega-3 polyunsaturated fatty acids in plasma after single-dose ingestion of echium oil. Concentration of (**A**) α-linolenic acid (ALA, 18:3*n*3), (**B**) stearidonic acid (SDA, 18:4*n*3), (**C**) eicosapentaenoic acid (EPA, 20:5*n*3), (**D**) docosapentaenoic acid (DPA*n*3, 22:5*n*3), (**E**) docosahexaenoic acid (DHA, 22:6*n*3), and (**F**) sum of omega-3 polyunsaturated fatty acids (Σ*n*3 PUFA) in plasma at baseline 0, and 2, 4, 6, 8, 24, 48, 72 h after single-dose ingestion of echium oil. Levels are shown as concentrations [μg/mL] and as relative concentrations (%) of total fatty acids. All data are shown as mean ± standard deviation. Levels of significance (for normally distributed variables: one-factorial ANOVAs with repeated measurements and t-tests for paired samples with Holm–Bonferroni correction; for not normally distributed variables: Friedman test followed by Dunn–Bonferroni post-hoc test) are indicated as follows: * *p* ≤ 0.05, # *p* ≤ 0.005, ‡ *p* ≤ 0.001. < LLOQ, below lower level of quantification.

**Table 1 nutrients-11-02346-t001:** Fatty acid pattern (concentration in g/100 g) of the echium oil used in the study^1^.

Fatty Acid	Common Name	Concentration (g/100 g)
16:0	Palmitic acid	6.95
16:1*n*7	Palmitoleic acid	0.0925
18:0	Stearic acid	3.62
18:1*n*9	Oleic acid	15.5
18:1*n*7	Vaccenic acid	0.910
18:2*n6*	Linoleic acid	15.2
18:3*n*6	γ-Linolenic acid	10.4
18:3*n*3	α-Linolenic acid	30.2
18:4*n*3	Stearidonic acid	11.8
20:0	Arachidic acid	0.0980
20:1*n*9	Gondoic acid	0.592
22:0	Behenic acid	0.117
22:1*n*9	Erucic acid	0.129
SFA		10.8
MUFA		17.3
PUFA		67.6
Σ*n*3 PUFA		42.0
Σ*n*6 PUFA		25.6

^1^ own analysis. *n*3: omega-3; *n*6: omega-6; MUFA: monounsaturated fatty acids; PUFA: polyunsaturated fatty acids; SFA: saturated fatty acids.

**Table 2 nutrients-11-02346-t002:** Baseline clinical, biochemical, and anthropometric parameters of study participants. Shown are mean ± SD (*n* = 12).

Parameter	mean	±	SD
Age (years)	24.6	±	2.43
Weight (kg)	84.5	±	8.79
BMI (kg/m^2^)	24.6	±	2.06
Sys BP (mmHg)	123	±	13.9
Dias BP (mmHg)	74.2	±	5.57
AST (U/l)	25.4	±	5.28
ALT (U/l)	23.0	±	13.5
GGT (U/l)	20.3	±	8.54
TC (mg/dl)	166	±	38.4

ALT: Alanine Aminotransferase; AST: Aspartate Aminotransferase; BMI: body mass index; dias BP: diastolic blood pressure; GGT: Gamma-glutamyl transpeptidase; HDL: high density lipoprotein; LDL: low density lipoprotein; SD: standard deviation; sys BP: systolic blood pressure; TC: total cholesterol; TG: triglycerides.

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
