# Peer review of "Single-Dose SDA-Rich Echium Oil Increases Plasma EPA, DPAn3, and DHA Concentrations"

_nutrients, 2019, doi:10.3390/nu11102346_

Round 1

Reviewer 1 Report

The article is well written, adequately explained, and the figure is demonstrative; however there is an intrinsic complexity of the topic that should be reduced to make it easier to the rider. I suggest to combine the numerous acronyms used in a table.

The side effects in the volunteers tested, that could even in the short time be occurred, are missing. I recommend to add or explain why not. 

At line 37, the meaning of LA and its synthetic formula are missing.

Ay line 195, a sentence ends with : and is followed by It with capital letter

Finally a question: how was determined the single dose of 26 g of echium oil? How much is the daily requirement of stearidonic acid, if any?

Reviewer 2 Report

The article by Greupner et al reports that single dose of 26g of  Echium oil increases plasma EPA and DPA concentrations and DHA to some extent in healthy men. The time course results are very interesting. The study did control the diet of the participants on the day before and during the testing period which is a strength. I have provided section wise comments below.

Abstract

The authors use both omega-3 and n3. Please use a consistent terminology. Please provide EPA, DPA and DHA full form as you have done for ALA and SDA in the abstract. Please proc=vide quantification of results. So at 72 hours the EPA, DPA and DHA levels were increased by what % compared to baseline? The phrase ‘slightly also’ does not sound grammatically correct. Please rephrase.

Introduction

The introduction states EPA and DHA have health benefits. Why has DPA been left out here? There is plenty of literature suggesting DPA has (unique) biological effects as well. Please add this in the introduction. Refer to- Kaur G, Cameron-Smith D, Garg M and Sinclair AJ. Docosapentaenoic acid (22:5n-3): a review of its biological effects. Prog Lipid Res 2011. 50 (1): 28-34. Kaur G, Guo XF, Sinclair AJ. Short update on docosapentaenoic acid: a bioactive long-chain n-3 fatty acid. Current opinion in clinical nutrition and metabolic care. 2016;19(2):88-91. Drouin G, Rioux V, Legrand P. The n-3 docosapentaenoic acid (DPA): A new player in the n-3 long chain polyunsaturated fatty acid family. Biochimie. 2019 Apr;159:36-48.

Methods

How did the authors decide on 12 participants? Please provide power calculations. Line 63 Define LO. Line 80 What does ‘small blood picture’ mean? Line 98 Please mention (table 2). Line 101 If analyte could not be quantified why is it included? It’s a bit unclear.

Discussion

Line 198 Was the FA composition of EO in Ref 45 and the current study exactly the same? Line207 Please add ‘after’ 8 and 72 hours. I think the discussion needs a paragraph on study limitations. Also the significance needs to come across more. Are you suggesting EO is a better/cheaper or more sustainable source for n3 supplementation? What do the authors mean by DHA conc was unsteady? It is significantly high at 72 hours so why is it unsteady?

Appendix

The recruitment section is repeated in Appendix and in Supplementary materials. Just supplementary should be enough
